# Continued value of the serum alpha-fetoprotein test in surveilling at-risk populations for hepatocellular carcinoma

Jihyun An[1], Ha Il Kim[2], Seheon Chang[3], Ju Hyun Shim[4,5]*

**1** Gastroenterology and Hepatology, Hanyang University College of Medicine, Guri, Korea,
**2** Gastroenterology, Kyung Hee University Hospital at Gangdong, Seoul, Korea, **3** Internal Medicine, Myongji Saint Mary's Hospital, Seoul, Korea, **4** Asan Liver Center, Asan Medical Center, University of Ulsan College of Medicine, Seoul, Korea, **5** Gastroenterology, Asan Medical Center, University of Ulsan College of Medicine, Seoul, Korea

☯ These authors contributed equally to this work.
* s5854@amc.seoul.kr

**Data Availability Statement:** All relevant data are within the manuscript and its Supporting Information files.

**Funding:** This study was supported by Basic Science Research Program through the National

## Abstract

### Backgrounds and aims

Because of the known limitations of ultrasonography (US) alone, we re-evaluated whether complimentary testing for serum alpha-fetoprotein (AFP) is helpful in surveilling for hepatocellular carcinoma (HCC) in high-risk populations.

### Methods

We included, from a hospital-based cancer registry, 1,776 asymptomatic adults who were surveilled biannually with the AFP test and US and eventually diagnosed with HCC between 2007 and 2015. Based on the screening results, these patients were divided into three groups: AFP (positive for AFP only; n = 298 [16.8%]), US (positive for US only; n = 978 [55.0%]), and AFP+US (positive for both; n = 500 [28.2%]). We compared the outcomes of the three groups, calculating the survival of the AFP group both as observed survival and as survival corrected for lead-time.

### Results

In terms of tumor-related factors, the separate AFP and US groups were more likely to have early stage HCC and to receive curative treatments than the combined AFP+US group (Ps<0.05). The AFP group had significantly better overall and cancer-specific survival than the AFP+US group after adjusting for covariates (adjusted hazard ratios [HRs] 0.68 and 0.62, respectively). In analyses correcting for lead-time in the AFP group (doubling time 120 days), the respective adjusted HRs for the AFP group were unchanged (0.74 and 0.67), but they were no longer significant after additional adjustment for tumor stage and curative treatment (0.87 and 0.81).

Research Foundation of Korea funded by the Ministry of Science and ICT (NRF-2017R1E1A1A01074298; Initials of the authors: JHS) and the research fund of Hanyang University (HY-201900000002619; Initials of the authors: JA). The funders had no role in study design, data collection and analysis, decision to publish, or preparation of the manuscript.

**Competing interests:** The authors have declared that no competing interests exist.

## Conclusions

HCC cases detected by the AFP test without abnormal ultrasonic findings appear to have better survival, possibly as a result of stage migration and the resulting cures. Complementary AFP surveillance, together with US, could be helpful for at-risk patients.

## Introduction

Hepatocellular carcinoma (HCC) is one of the fastest growing causes of cancer death globally. [1] It has a reputation as a rapidly progressive cancer that is almost invariably fatal, with 3-year survival of less than 30%.[2] The high case-fatality ratio of HCC may be attributed in part of its vague and nonspecific symptoms, which usually appear when the disease has reached an advanced stage. However, a considerable improvement in survival has been observed in patients who have early-detected HCC and thus receive potentially curative treatment.[3]

Surveillance for HCC is thought to be a way to detect lesions at an earlier stage and improve clinical outcomes in asymptomatic at-risk populations.[4,5] Currently, ultrasonography (US) is regarded as the backbone of screening for HCC, but it has practical limitations in terms of high operator-dependency and variable sensitivity. A recent meta-analysis highlights suboptimal (<50%) sensitivity of US for detection of HCC at an early stage, although the sensitivity and specificity of US in detecting HCC of any stage exceeds 90%.[6] The serum alpha-fetoprotein (AFP) assay is the only serological screening tool for early detection of HCC that reliably meets the final 5-phases criterion for possibly reducing the population disease burden listed by the Early Detection Research Network (EDRN), an initiative of the National Cancer Institute (NCI).[7] However, the use of AFP in surveillance is subject to ongoing debate, even as an adjunct to USG, due to issues about cost-effectiveness.[8–10] There have been false-positive results for HCC detection due to AFP elevation encountered in chronic liver disease, and false-negative results in HCCs not secreting AFP.[11] Unfortunately, there are no robust or promising next-generation biomarkers available for clinical use in screening or diagnostic systems.

Both the American Association for the Study of Liver Diseases (AASLD) and the National Comprehensive Cancer Network (NCCN) guidelines for HCC, newly updated in 2017, reintroduce serum AFP testing as an additional surveillance method because of its potential benefit for patients with HCCs secreting AFP.[12,13] Recent interesting efforts in the U.S. to develop a computer model for detecting HCC also included serum AFP measurement in the prediction calculator.[14,15]

We performed the present study to reappraise the methodological role of serum AFP, the oldest, and only available, oncomarker for HCC, for early detection of tumors. To this end, we examined the serum AFP values and ultrasonic results obtained at the time of HCC detection in a hospital-based cohort of patients undergoing regular surveillance in the absence of cancer-related symptoms, and then assessed survival outcomes according to diagnostic status.

## Methods

### Study subjects and design of the experimental groups

A study population consisting of 9,615 patients aged ≥20 years with HCC primarily diagnosed and treated between January 2007 and December 2015 was retrospectively constructed from a prospective hospital-based registry maintaining data on all new cases of cancer in Asan Medical Center, which is a part of the National Cancer Registration Program. This study was

approved by the institutional review board (IRB) of the Asan Medical Center, Seoul, Republic of Korea. (IRB No.2017-0029), and informed consent was waived by IRB.

Since the objective of the study was to evaluate the surveillance effect of the serum AFP test on a background of US examination, 7,766 patients were excluded as follows: (1) 1,904 patients who initially had cancer-related symptoms such as ascites, abdominal pain, fever, jaundice and constitutional syndromes (i.e., weight loss, malaise, and anorexia)[16]; (2) 5,786 who did not undergo biannual regular surveillance tests for HCC comprising both abdominal US and serum AFP, as formally recommended by Korean practice guidelines[17]; and (3) 76 who had concurrent non-HCC malignancies.

Among the 1,849 asymptomatic patients who underwent surveillance tests for HCC at 6-month intervals, there were (1) 47 patients for whom there was a time interval exceeding a month between the two screening tests and the subsequent diagnosis of HCC; (2) 14 who had any abnormal surveillance results that were not evaluated by further diagnostic tests; (3) 12 whose tumor was incidentally identified by other test modalities such as computed tomography.

Accordingly, a total of 1,776 patients whose HCC was detected during bi-annual concurrent ultrasound and AFP surveillance were included in the final analysis. The patients were classified into three groups according to the results of screening prior to confirmation of HCC as follows: an AFP group: patients with high serum AFP test results (≥20 ng/ml) but no focal lesions on ultrasonography; a US group: patients with suspected malignant lesion(s) on ultrasonography but normal AFP levels; and an AFP+US group: patients with positive findings in both tests (Fig 1)

## Screening and diagnostic approaches

Serum AFP was measured using an immunoradiometric assay (RIA-gnost AFP, Cis-Bio International, Schering, Switzerland) based on the principle of the sandwich assay with $I_{125}$-labeled

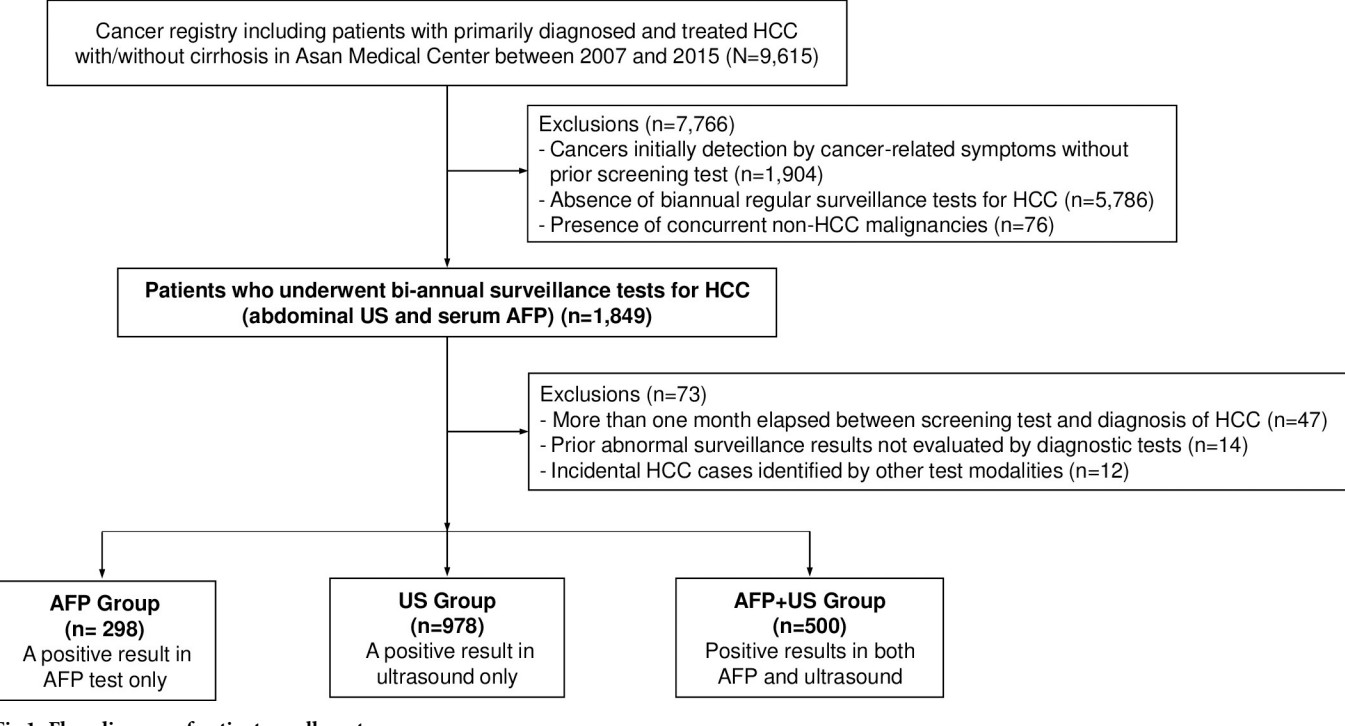

**Fig 1. Flow diagram of patient enrollment.**

anti-AFP monoclonal antibody, or a commercial enzyme immunoassay (Abbott AFP-EIA; Abbott Laboratories, North Chicago, IL). There is known to be good agreement between the results obtained by the two methods.[18] Ultrasonography was performed by licensed radiologists experienced in hepatobiliary ultrasound using real-time scanners. The diagnostic workup for HCC was initiated when AFP levels were elevated and/or a suspicious lesion(s) was observed on US. [19,20] The blood concentration of AFP used to identify a "screen-positive" result was ≥20 ng/ml, the typical criterion used in many surveillance studies and proven to be optimal for HCC screening.[19–21].

A diagnosis of HCC was established from either pathological (AFP *vs*. US *vs*. AFP+US group, 47.0% [n = 140] *vs*. 46.0% [n = 450] *vs*. 46.4% [n = 232], respectively) or radiological findings (AFP *vs*. US *vs*. AFP+US group, 53.0% [n = 158] vs. 54.0% [n = 528] vs. 53.6% [n = 268], respectively) in accordance with the international guidelines.[10,12] HCC stage at diagnosis was classified by the American Joint Committee on Cancer (AJCC) staging systems and Barcelona Clinic Liver Cancer (BCLC).[10,22] For each patient, the therapeutic modality for HCC was decided in a hierarchical manner according to efficacy in lengthening life based on tumor stage when this was feasible.

Detailed information on demographics, clinical data, tumors and survival outcomes were extracted from inpatient and outpatient medical records using the anonymized clinical database system of our institution (ABLE)[23] and the database of the National Population Registry of the Korea National Statistical Office using the unique personal identification numbers of the patients.

## Statistical analysis of observational data

For comparisons of the three groups, continuous variables were analyzed by one-way analysis of variance (ANOVA) and non-parametric testing, including the Kruskal-Wallis test, whilst categorical variables were assessed using the chi-square test and Fisher's exact test. Clinical variables associated with the "AFP group" were analyzed by the logistic regression method. Kaplan-Meier analysis was used to illustrate and compare overall survival across the groups, defined as the interval between the dates of HCC diagnosis and deaths from any cause. Death, survival, and follow-up data were fully accessible through the registry of our institution and were collected up to December 31, 2016. Cancer-specific mortality, in which deaths due to HCC progression were regarded as outcomes, was also measured. A Cox proportional hazard model with backward elimination was used to identify the independent characteristics of the groups associated with overall survival and HCC-specific death. Potential confounders with $P < 0.10$ among the demographic and hepatic variables in the univariate model, including age, sex, cause of chronic liver disease, method of HCC detection, body mass index, family history of HCC, positive history of alcohol and smoking, diabetes, hypertension, liver cirrhosis, ascites, model for end-stage liver disease (MELD) score, platelet counts, and infiltrative type of HCC, were used as input variables in the multivariate analysis. We hypothesized that any potential effect of the surveillance results on outcome would be due to stage migration and/or receipt of more curative treatment. Therefore, we examined changes in the parameter estimate of surveillance in the full model before (Model 1) and after (Model 2) adding these two potential explanatory variables. To reduce the impact of potential confounding effects in the AFP group and the AFP+US group, rigorous adjustment was made for significant differences in baseline characteristics in Model 1 and Model 2 by propensity score-based matching. Propensity scores were matched for the two groups based on differences of ±0.05 in the scores. Differences in overall mortality and cancer-specific mortality between the matched groups were compared using Cox regression models, with robust standard errors that accounted for the clustering of matched pairs.[24] $P < 0.05$ was considered statistically significant. All statistical analyses were performed using R package *MatchIT*.[25]

Because survival in the AFP group may be related to diagnoses made before the lesions were detected by US, we calculated lead times for the AFP group using Schwartz's formula. [26] Tumor volume doubling time was based on the value given in previous reports.[27–31] The estimated lead time for the AFP group was subtracted from their observed survival. If the value became negative, we attributed a survival (deceased patients) or a follow-up (surviving patients) of 1 day. The survival of the AFP group (corrected for the estimated lead time) was compared with the observed survival of the US and AFP+US groups. The adjusted hazard ratios (HRs) for corrected survival by the detection methods were also calculated using Cox multivariate stepwise regression analysis. The length-time bias was also adjusted using various tumor volume-doubling times from 90 to 150 days, which might represent tumors with various growth rates, for the calculation of the lead times.[27] A two-tailed $P$-value <0.05 was considered statistically significant. Statistical analyses were performed with R software version 3.1.1.

## Results

### Demographic and tumor characteristics of the patients

The median age of the 1,776 patients was 57 years (interquartile range [IQR], 51–64 years), and most of the patients were male (77.5%). Liver cirrhosis was present in 86.5% of the patients (1,536 out of 1,776), and 1,615 (90.9%) were in Child–Pugh class A. There were 46 (2.6%) patients with preclinical ascites and no related symptoms. When the patients were stratified based on the results of the screening tests, 298 (16.8%), 978 (55.1%), and 500 (28.2%) were assigned to the AFP, US, and AFP+US groups, respectively. Table 1 presents the baseline characteristics of the three groups in the screening period. The largest proportions of females and never drinkers were observed in the AFP group ($Ps$<0.001 for both). In terms of cause of liver disease, hepatitis B virus (HBV) infection was the most common risk factor for HCC in the entire cohort (82.1%). In terms of laboratory findings, mean hepatic inflammatory parameters such as aspartate transaminase and alanine transaminase levels were comparable across the groups ($P$ = 0.162 and $P$ = 0.378, respectively). However, liver cirrhosis was more common in the AFP group, as was more advanced liver dysfunction (i.e., Child-Pugh class B) ($Ps$<0.05). Mean platelet count, representing underlying portal hypertension, decreased from the AFP +US group through the US and AFP groups ($Ps$<0.001).

The median interval between screening test and diagnosis of HCC was 2.0 weeks (IQR 1.7– 2.2 weeks) with similar data across the three groups ($P$ = 0.445; Table 2). The initial findings for tumor stage and therapeutic modality in the three screening groups are shown in Table 2. The median maximal tumor diameter was 2.4 cm (IQR, 1.7–3.5 cm), and 7.3% of the patients had three or more tumors. The infiltrative subtype of HCC that is difficult to recognize on early imaging was least common in the US group (1.6%), and present in 5.0% of the AFP group ($P$<0.001). Multiple tumors and larger tumors were most frequent in the AFP+US group, with similar numbers in the AFP and US groups ($Ps$<0.001 for all). While the proportion of patients with early HCC (i.e., very early and early stage HCC based on the BCLC classification) was highest in the AFP group (83.2% $vs$. 67.6%, $P$<0.001), more advanced tumors were most frequent in the AFP+US group.

Surgical resection and chemoembolization were the most frequent initial methods used for treating HCC in all the groups. Primary liver transplantation was performed in 15 (5.0%), 22 (2.3%), and 10 (2.0%) patients, respectively, in the AFP, US, and AFP+US groups. Curative treatments such as resection, transplantation and local ablative therapies were more often performed in the AFP group and US group than in the AFP+US group (60.1% $vs$. 63.1% $vs$. 56.4%, $P$ = 0.044; Table 2)

**Table 1. Demographic characteristics at the time of screening of 1,776 patients with hepatocellular carcinoma.**

| Variable | AFP group (n = 298) | US group (n = 978) | AFP+US group (n = 500) | *P* value |
|---|---|---|---|---|
| *Demographic factor* | | | | |
| Male sex | 200 (67.1%) | 817 (83.5%) | 360 (72.0%) | <0.001 |
| Age (years) | 56 (50–64) | 58 (52–64) | 56 (50–63) | 0.001 |
| Body mass index (kg/m$^2$) | 24.6 (22.9–26.5) | 24.8 (22.9–26.7) | 24.3 (22.3–26.1) | 0.018 |
| Alcohol consumption | | | | <0.001 |
| None | 150 (50.3%) | 379 (38.8%) | 232 (46.4%) | |
| Former | 109 (36.3%) | 394 (40.2%) | 182 (36.4%) | |
| Current | 39 (13.1%) | 205 (21.0%) | 86 (17.2%) | |
| Smoking habitus | | | | <0.001 |
| None | 163 (54.7%) | 414 (42.3%) | 255 (51.0%) | |
| Former | 104 (34.9%) | 382 (39.1%) | 149 (29.8%) | |
| Current | 31 (10.4%) | 182 (18.6%) | 96 (19.2%) | |
| Family history of HCC | 42 (14.1%) | 120 (12.3%) | 75 (15.0%) | 0.316 |
| Hypertension | 75 (25.2%) | 272 (27.8%) | 123 (24.6%) | 0.356 |
| Diabetes mellitus | 53 (17.8%) | 224 (22.9%) | 65 (13.0%) | <0.001 |
| *Liver disease-related factor* | | | | |
| Hepatitis B virus infection | 239 (80.2%) | 800 (81.8%) | 419 (83.8%) | 0.412 |
| Hepatitis C virus infection | 36 (12.1%) | 77 (7.9%) | 49 (9.8%) | 0.072 |
| Liver cirrhosis | 275 (92.3%) | 833 (85.2%) | 428 (85.6%) | 0.006 |
| Ascites | 13 (4.4%) | 22 (2.2%) | 11 (2.2%) | 0.107 |
| Serum AST (IU/L) | 41 (30–61) | 41 (29–59) | 38 (28–57) | 0.162 |
| Serum ALT (IU/L) | 36 (24–55) | 34 (22–50) | 33 (23–47) | 0.378 |
| Platelet count (X10$^3$/mm$^3$) | 113 (83–151) | 127 (91–166) | 134 (90–170) | <0.001 |
| Serum albumin (g/dl) | 3.8 (3.4–4.1) | 3.9 (3.5–4.2) | 3.9 (3.5–4.2) | 0.280 |
| Serum bilirubin (mg/dl) | 0.9 (0.7–1.3) | 0.8 (0.6–1.2) | 0.9 (0.6–1.2) | 0.119 |
| International normalized ratio (INR) | 1.07 (1.03–1.17) | 1.06 (1.02–1.15) | 1.07 (1.02–1.15) | 0.065 |
| Serum creatinine (mg/dl) | 0.80 (0.70–0.94) | 0.84 (0.71–0.97) | 0.80 (0.70–0.96) | 0.849 |
| Child-Pugh class | | | | 0.005 |
| class A | 256 (85.9%) | 898 (91.8%) | 460 (92.0%) | |
| class B | 42 (14.1%) | 80 (8.2%) | 40 (8.0%) | |
| MELD score | 8 (7–9) | 8 (7–9) | 7 (7–9) | 0.133 |
| *Serum alpha-fetoprotein (AFP)* | | | | |
| Median, IQR (ng/ml) | 134.3 (54.4–355.8) | 5.3 (2.9–9.1) | 133.8 (45.8–608.4) | <0.001 |
| Median, IQR (log$_{10}$ng/ml) | 2.13 (1.74–2.55) | 0.72 (0.46–0.96) | 2.13 (1.66–2.88) | |
| Mean, SD (ng/ml) | 357.9 ± 536.1 | 6.5 ± 4.7 | 553.3 ± 984.1 | |
| Mean, SD (log$_{10}$ng/ml) | 2.19 ± 0.55 | 0.70 ± 0.34 | 2.34 ± 0.87 | |
| Method of measurement | | | | 0.852 |
| Enzyme immunoassay (EIA) | 148 (49.7%) | 504 (51.5%) | 256 (51.2%) | |
| Radioimmunoassay (RIA) | 150 (50.3%) | 474 (48.5%) | 244 (48.8%) | |

Data are presented as number (%) or median (interquartile range), unless otherwise indicated.

AFP, alpha-fetoprotein; US, ultrasonography; HCC, hepatocellular carcinoma; AST, aspartate transaminase; ALT, alanine transaminase; MELD, model for end-stage liver disease; IQR, interquartile range; SD, standard **deviation**.

## Demographic and hepatic factors associated with the AFP group

Table 3 shows the relationships between demographic and liver disease-related parameters and the AFP group. In univariate analysis, female gender, habitus of alcohol drinking and

**Table 2. Differences in initial tumor and treatment factors according to the positive results in the screening tests.**

| Variable | AFP group (n = 298) | US group (n = 978) | AFP+US group (n = 500) | P value |
|---|---|---|---|---|
| *Interval between screening test and diagnosis of HCC* | | | | |
| Time interval (weeks) | 2.0 (1.7–2.3) | 1.9 (1.6–2.2) | 2.0 (1.8–2.3) | 0.445 |
| *Tumor characteristic* | | | | |
| Number of tumors | | | | 0.001 |
| 1 | 215 (72.1%) | 717 (73.3%) | 323 (64.6%) | |
| 2–3 | 69 (23.2%) | 199 (20.3%) | 123 (24.6%) | |
| >3 | 14 (4.7%) | 62 (6.4%) | 54 (10.8%) | |
| Maximal tumor size (cm) | 2.0 (1.5–3.0) | 2.3 (1.7–3.3) | 3.0 (2.0–5.0) | <0.001 |
| Infiltrative type of HCC | 15 (5.0%) | 16 (1.6%) | 28 (5.6%) | <0.001 |
| Vascular invasion | 25 (8.4%) | 53 (5.4%) | 60 (12.8%) | <0.001 |
| Extra-hepatic metastasis | 8 (2.7%) | 19 (1.9%) | 20 (4.0%) | 0.066 |
| *Cancer stage* | | | | |
| BCLC staging | | | | <0.001 |
| Stage 0 | 94 (31.5%) | 232 (23.7%) | 82 (16.4%) | |
| Stage A | 154 (51.7%) | 574 (58.7%) | 260 (52.0%) | |
| Stage B | 19 (6.4%) | 105 (10.7%) | 88 (17.6%) | |
| Stage C | 31 (10.4%) | 67 (6.9%) | 70 (14.0%) | |
| AJCC staging | | | | <0.001 |
| Stage IA | 119 (39.9%) | 305 (31.2%) | 112 (22.4%) | |
| Stage IB | 79 (26.5%) | 367 (37.5%) | 174 (34.8%) | |
| Stage II | 72 (24.2%) | 239 (24.4%) | 130 (26.0%) | |
| Stage IIIA | 7 (2.3%) | 20 (2.0%) | 32 (6.4%) | |
| Stage IIIB | 13 (4.4%) | 28 (2.9%) | 32 (6.4%) | |
| Stage IVA-IVB | 8 (2.7%) | 19 (1.9%) | 20 (4.0%) | |
| *Initial anti-cancer treatment* | | | | |
| Surgical resection | 123 (41.3%) | 415 (42.4%) | 218 (43.6%) | 0.044* |
| Local ablation therapy | 41 (13.8%) | 180 (18.4%) | 54 (10.8%) | |
| Liver transplantation | 15 (5.0%) | 22 (2.3%) | 10 (2.0%) | |
| Transarterial chemoembolization | 111 (37.2%) | 349 (35.7%) | 204 (40.8%) | |
| Radiotherapy | 2 (0.7%) | 3 (0.3%) | 3 (0.6%) | |
| Systemic chemotherapy | 2 (0.7%) | 0 (0%) | 3 (0.6%) | |
| Conservative management | 4 (1.3%) | 9 (0.9%) | 8 (1.6%) | |

Data are presented as number (percentage) or median (interquartile range)

*P values for curative (i.e., surgical resection, local ablation, and liver transplantation) versus non-curative treatment.

AFP, alpha-fetoprotein; US, ultrasonography; HCC, hepatocellular carcinoma; BCLC, Barcelona Clinic Liver Cancer; AJCC, American Joint Committee on Cancer.

smoking, presence of cirrhosis or ascites, Child-Pugh class B liver function, and lower platelet count were significantly associated with the AFP group. After adjusting for the confounding covariates, female gender (odds ratio [OR], 0.53; 95% confidence interval [CI], 0.40–0.70) was significantly associated with the AFP group. In terms of hepatic variables, cirrhosis of the liver (OR 1.91; 95% CI 1.21–3.00), and Child-Pugh class B liver function (OR 1.69; CI 1.16–2.48) were independent factors related to the AFP group. We investigated the skewed effect potentially induced by AFP elevation and directly associated with liver cirrhosis *per se*, and then found that the mean log-transformed AFP level was comparable in the cirrhotic- and non-cirrhotic patients (1.42 ± 0.95 $\log_{10}$ng/ml vs. 1.36 ± 1.15 $\log_{10}$ng/ml, $P$ = 0.427).

**Table 3. Baseline parameters related to HCC patients with positive results only for AFP screening.**

| Variable | Univariate analysis | | | Multivariate analysis | | |
|---|---|---|---|---|---|---|
| | OR | 95% CI | *P* value | OR | 95% CI | *P* value |
| Male | 0.52 | 0.40–0.69 | <0.001 | 0.53 | 0.40–0.70 | <0.001 |
| Age ≥50 years | 0.79 | 0.58–1.07 | 0.124 | - | - | - |
| Body mass index >25 kg/m$^2$ | 0.92 | 0.72–1.19 | 0.533 | - | - | - |
| Alcohol drinking | 0.70 | 0.54–0.89 | 0.004 | 0.97 | 0.70–1.33 | 0.828 |
| Smoking habitus | 0.69 | 0.53–0.88 | 0.003 | 0.95 | 0.70–1.29 | 0.724 |
| Family history of HCC | 1.08 | 0.75–1.55 | 0.677 | - | - | - |
| HBV infection | 0.86 | 0.63–1.18 | 0.350 | - | - | - |
| HCV infection | 1.47 | 1.00–2.19 | 0.053 | 1.28 | 0.85–1.91 | 0.233 |
| Serum AST >ULN | 1.10 | 0.86–1.41 | 0.458 | - | - | - |
| Serum ALT >ULN | 1.25 | 0.97–1.60 | 0.089 | 1.27 | 0.98–1.63 | 0.071- |
| Liver cirrhosis | 2.06 | 1.31–3.22 | 0.002 | 1.91 | 1.21–3.00 | 0.005 |
| Ascites | 2.00 | 1.04–3.84 | 0.038 | 1.31 | 0.62–2.77 | 0.477 |
| Child-Pugh class B | 1.86 | 1.28–2.70 | 0.001 | 1.69 | 1.16–2.48 | 0.007 |
| MELD score | 1.05 | 1.00–1.11 | 0.050 | 0.99 | 0.92–1.06 | 0.731 |
| Platelet count <100k/mm$^3$ | 1.52 | 1.18–1.96 | 0.001 | 1.20 | 0.91–1.58 | 0.202 |

HCC, hepatocellular carcinoma; AFP, alpha-fetoprotein; OR, odds ratio; CI, confidence interval; HBV, hepatitis B virus; HCV, hepatitis C virus; AST, aspartate transaminase; ULN, upper limit of normal; ALT, alanine transaminase; MELD, model for end-stage liver disease.

In terms of tumor factors, about two thirds of the HCC tumors (68.5%) in the AFP group were in previously established blind spots of the US test, including hepatic dome, area under the rib, caudate lobe, and tip of the lateral segment of the left lobe.

## Association between screening results and survival outcomes

During a median follow-up of 3.09 years (IQR 1.60–5.13 years), 539 patients (30.3%) died due to all causes: 88 (29.5%) in the AFP group, 253 (25.9%) in the US group, and 198 (39.6%) in the AFP+US group. HCC-related deaths occurred in 370 patients (20.8%) with 63 (20.1%), 162 (16.6%), and 148 (29.6%) in the respective groups. The 5-year cumulative rates of overall and cancer-specific survival estimated by the Kaplan-Meier method were 64.8%, 69.9%, and 55.5%, and 74.8%, 79.2% and 64.2%, respectively, in the corresponding groups. There was a significant difference across the groups in both overall and cancer-specific survival (*P*<0.001 and *P*<0.001, respectively, by the log-rank test; Fig 2A and 2B).

Univariate analyses showed that the HRs of overall and cancer-specific survival relative to the AFP+US group were 0.65 and 0.59 for the AFP group, and 0.54 and 0.46 for the US group, respectively (all *Ps*≤0.001). Other significant variables related to time-dependent outcomes are presented in Table 4. These factors were subsequently assessed in a multivariate proportional hazard regression model (Model 1; Table 4). To evaluate the potential effect of AFP screening due to stage migration and subsequent receipt of more-curative treatment, BCLC stage and primary treatment were added to this model (Model 2; Table 4). In model 1, the AFP and US groups had multivariate HRs of 0.60 (CI 0.47–0.78) and 0.53 (CI 0.43–0.64), respectively, for overall death relative to the AFP+US group (*Ps*<0.001, respectively). In model 2, the adjusted HRs of the AFP and US groups compared with the AFP+US group for all-cause death were 0.68 (CI 0.52–0.88; *P* = 0.003) and 0.57 (CI 0.47–0.69; *P*<0.001), respectively. In terms of HCC-related mortality, the lower HRs in the AFP group than in the AFP+US group were also significant, as shown in Table 4 (adjusted HR [95% CI], 0.55 [0.41–0.75], *P*<0.001 in Model 1,

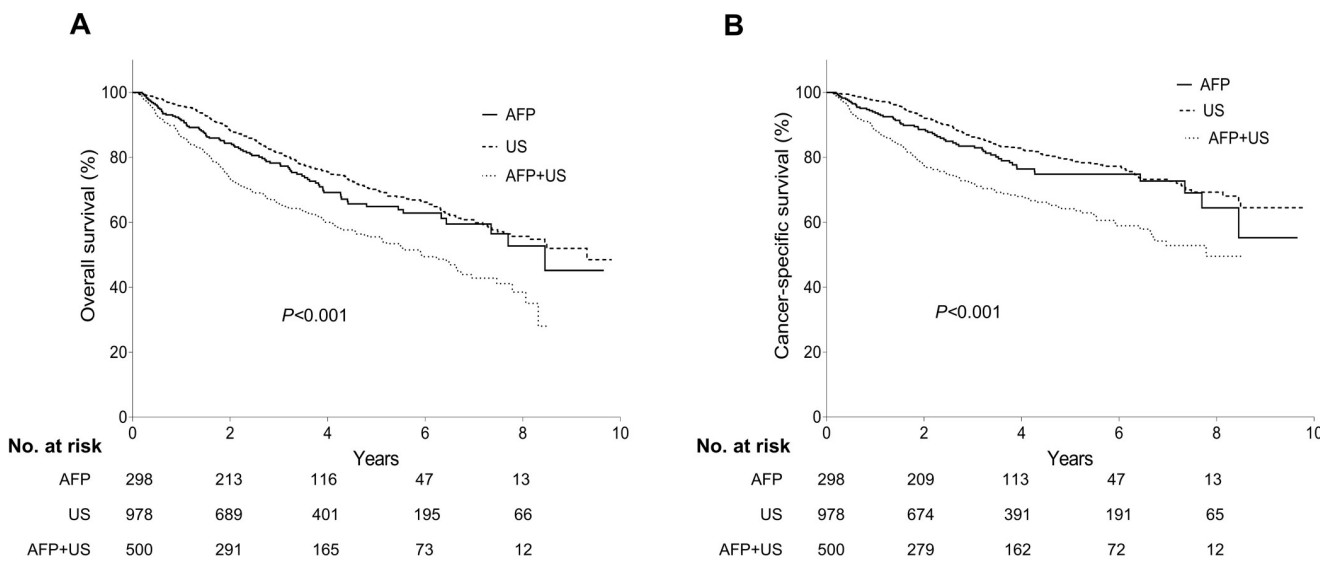

**Fig 2.** Associations between methods of HCC detection and (A) overall mortality, and (B) cancer-specific mortality. Overall and cancer-specific mortalities were significantly different across the groups (P<0.001 and P<0.001, respectively by the log-rank test).

**Table 4. Independent pre- and post-screening parameters related to survival in the entire HCC cohort (based on the BCLC system).**

| Variable | Overall morality | | | | | | Cancer-specific mortality | | | | | |
|---|---|---|---|---|---|---|---|---|---|---|---|---|
| | Model 1 | | | Model 2* | | | Model 1 | | | Model 2* | | |
| | HR | 95% CI | *P* | HR | 95% CI | *P* | HR | 95% CI | *P* | HR | 95% CI | *P* |
| **Group** | | | | | | | | | | | | |
| AFP+US group | 1 | | | | | | 1 | | | 1 | | |
| AFP group | 0.60 | 0.47–0.78 | <0.001 | 0.68 | 0.52–0.88 | 0.003 | 0.55 | 0.41–0.75 | <0.001 | 0.62 | 0.46–0.84 | 0.002 |
| US group | 0.53 | 0.43–0.64 | <0.001 | 0.57 | 0.47–0.69 | <0.001 | 0.46 | 0.37–0.58 | <0.001 | 0.50 | 0.40–0.63 | <0.001 |
| **Male sex** | - | - | - | - | - | - | 1.57 | 1.19–2.07 | 0.001 | 1.51 | 1.14–1.99 | 0.004 |
| **Diabetes** | 1.21 | 0.98–1.48 | 0.074 | 1.14 | 0.93–1.40 | 0.223 | - | - | - | - | - | - |
| **Positive history of alcohol consumption** | 1.37 | 1.15–1.63 | 0.001 | 1.28 | 1.08–1.53 | 0.005 | 1.21 | 0.96–1.53 | 0.114 | 1.18 | 0.93–1.50 | 0.163 |
| **HBV infection** | 0.61 | 0.47–0.78 | <0.001 | 0.56 | 0.46–0.69 | <0.001 | 0.68 | 0.53–0.87 | 0.003 | 0.79 | 0.61–1.01 | 0.063 |
| **HCV infection** | 1.31 | 0.96–1.81 | 0.094 | 1.24 | 0.90–1.70 | 0.183 | 1.17 | 0.77–1.80 | 0.459 | 1.08 | 0.71–1.66 | 0.722 |
| **Liver cirrhosis** | 1.11 | 0.80–1.53 | 0.540 | 0.93 | 0.67–1.30 | 0.685 | - | - | - | - | - | - |
| **Ascites** | 2.36 | 1.59–3.49 | <0.001 | 1.99 | 1.33–2.98 | 0.001 | 2.61 | 1.58–4.31 | <0.001 | 2.25 | 1.37–3.71 | 0.001 |
| **MELD score** | 1.08 | 1.05–1.12 | <0.001 | 1.06 | 1.03–1.10 | <0.001 | 1.05 | 1.01–1.10 | 0.010 | 1.01 | 0.97–1.06 | 0.525 |
| **Platelet count <100k/mm³** | 1.39 | 1.16–1.67 | <0.001 | 1.15 | 0.95–1.38 | 0.154 | 1.19 | 0.94–1.50 | 0.141 | 0.95 | 0.75–1.20 | 0.657 |
| **Infiltrative type of HCC** | 5.95 | 4.35–8.14 | <0.001 | 2.14 | 1.51–3.04 | <0.001 | 6.83 | 4.88–9.56 | <0.001 | 2.06 | 1.41–3.00 | <0.001 |
| **BCLC stage** | | | | | | | | | | | | |
| Stage 0 | | | | 1 | | | | | | 1 | | |
| Stage A | | | | 1.38 | 1.07–1.79 | 0.014 | | | | 1.77 | 1.23–2.53 | 0.002 |
| **Stage B** | | | | 1.85 | 1.35–2.52 | <0.001 | | | | 2.71 | 1.80–4.06 | <0.001 |
| Stage C | | | | 4.77 | 3.47–6.56 | <0.001 | | | | 8.60 | 5.74–12.88 | <0.001 |
| **Curative treatments** | | | | 0.34 | 0.28–0.41 | <0.001 | | | | 0.31 | 0.25–0.39 | <0.001 |

*Adjusted for BCLC stage, receipt of curative treatment, and all variables in Model 1.

HCC, hepatocellular carcinoma; BCLC, Barcelona clinic liver cancer; HR, hazard ratio; CI, confidence interval; AFP, alpha-fetoprotein; US, ultrasonography; HBV, hepatitis B virus; HCV, hepatitis C virus; MELD, model for end-stage liver disease.

and HR 0.62 [0.46–0.84], *P* = 0.002 in Model 2). The same trends were observed when using the AJCC staging system, as shown in S1 Table. In addition, these results were recapitulated after PS matching between the AFP and AFP+US groups (S2 Table and S1 and S2 Figs).

## Lead-time correction model

Because the earlier recognition in the preclinical course of disease due to AFP screening appeared to contribute to the longer survival observed in the AFP group, we corrected for lead-time in that group. When we used 120 days as the assumed tumor doubling time to correct for lead-time bias, overall survival and cancer-specific mortality outcomes were significantly better in the AFP group than in the AFP+US group (unadjusted HR [95% CI], 0.75 [0.58–0.96] and 0.67 [0.49–0.90], respectively, all *Ps*<0.001, S3 Fig and S3 Table). After adjusting for other confounders (model 1), the differences in the corresponding time-dependent outcomes between the AFP and AFP+US groups corrected for estimated lead-times remained significant (adjusted HR [95% CI], 0.74 [0.57–0.95], *P* = 0.019 and 0.67 [0.50–0.90], *P* = 0.009, respectively). However, the survival after correction for lead-time in the AFP group, together with adjustment for variables related to initial BCLC stage of tumor and its treatment, revealed comparable benefits in the AFP and AFP+US groups regarding both overall and cancer deaths (adjusted HR [95% CI], 0.87 [0.67–1.12], *P* = 0.276 and 0.81 [0.60–1.10], *P* = 0.178, respectively). All the outcomes remained similar when estimated tumor doubling times of 90 or 150 days were used, as shown in S3 Table, and similar results were obtained when the analysis was based on the AJCC staging system (S4 Table).

## Analysis of the cirrhotic subset

Analysis of the cirrhotic cohort (n = 1,536) showed that the significantly lower overall and cancer-specific mortality risks of the AFP group relative to the AFP+US group were maintained, with adjusted HRs of 0.67 (0.51–0.87; *P* = 0.003) and 0.59 (0.43–0.82; *P* = 0.002), respectively, after adjustment of BCLC stage and curative treatment options (S4 Fig and S5 Table). Survival after correction for lead-time in the AFP group also revealed similar trends in patients with liver cancer as were observed in the entire cohort (S6 Table).

## Discussion

In the present study, retrospective evaluation of data for preclinical HCC revealed that approximately 17% of asymptomatic patients with HCC were diagnosed on the basis of a preceding elevation of AFP, with no suspected lesions on ultrasonic images. We conclude that the serum AFP test can play a complementary role in the early detection of HCC even when there is regular US-based surveillance. This route to diagnosis was more often observed in cirrhotic patients, and, importantly, such patients had better survival outcomes than those whose HCC was detected by both the serological and radiological tests. The trend for our findings by the two competing endpoints of all-cause and cancer-specific mortality to coincide adds to the evidence that the AFP test helps early detection of HCC. Because of the finding that the difference in survival between the patients identified by AFP with and without abnormal US findings was maintained after correcting for lead time combined with confounding adjustment, we suggest that the observed survival benefit was due to downward stage migration followed by more frequent receipt of potentially curative treatment.

 The clinical effectiveness of an HCC screening program in at-risk individuals relies on early diagnosis, provided that effective treatments are available.[4,5] Consequently, an increased proportion of successful treatments and a reduction in the mortality of surveilled patients should be the final results. While US is a well-accepted modality for HCC surveillance, the

interpretation of US can be difficult due to the increased liver nodularity and parenchymal heterogeneity as cirrhosis progresses, which hampers the identification of nodules.[19,32] There is currently no concrete evidence supporting the use of CT or MRI as part of a routine surveillance strategy for detecting HCC. In addition, the high costs and potential harm related to contrast-related injury and radiation exposure associated with these tests make them poor candidates for surveillance tests in most clinical settings.[33] In our regular surveillance series, nearly 83.2% of the cancers positive for AFP screening alone were detected at early, potentially curable, stages, as were the HCC nodules primarily identified on US that did not produce AFP and had more indolent phenotypes. In contrast, over 68.4% of the cases that gave screen-positive results in both tests were beyond the early stages and produced more AFP.

There has been controversy about the complementary use of AFP testing as a biomarker for HCC surveillance.[8,10] A stage-specific meta-analysis of 13 prospective studies found only a minimal effect of adding AFP measurement on the sensitivity of detection of early HCC.[9] However, inter-study heterogeneity in the included individuals, in the AFP cut-offs, and in the reference imaging make these findings unreliable. In addition to the survival benefit of AFP testing in our present work, recent investigations of the accuracy of HCC surveillance techniques have reported greater sensitivity of US and AFP for early stage HCC in cirrhotics compared to US alone,[6] and a lower false-positive rate for AFP than for US.[34] Indeed, the most recently updated oncology and hepatology guidelines (re)introduce AFP.[12,13]

In order to establish mandatory indications of AFP for HCC detection, we need to determine the potential predictors of HCC as detected by "elevated AFP data without pathologic ultrasonic information". In our series, liver cirrhosis, which is usually accompanied by portal hypertension, was major predictor requiring complementary AFP screening. For our AFP-secreting HCCs in our cirrhotic subset, the actual outcomes were better when they were only serologically detectable outside the ultrasonic detection range in which cirrhotic patients, albeit more cancer-susceptible, were more often found. Our evidence-based data support the statement by the American College of Gastroenterology that diagnostic examination should be done in cirrhotics with an elevated or rising AFP, even in the absence of abnormal findings on US. [35] Our observations could provide a rationale for the clinical use of the AFP serotest coupled with US screening, particularly in the cirrhotic subjects, in order to provide life-saving or life-extending opportunities to impending HCC patients.

There are potential limitations to our study. A complimentary role of AFP testing in HCC surveillance would ideally be evaluated by a randomized trial in a high-risk population, especially to assess the consequences of falsely elevated AFP. Although a formal economic analysis will be needed to justify offering AFP testing as a concrete complimentary option, we believe that developed societies could afford the cost of the false-positive AFP results added to the 90% true positives at least in cirrhotic patients in whom the sensitivity of US for early tumors is disappointing and the risk of HCC is highest,[36] if that improved survival, as in our findings. Second, length-time bias is generally recognized as important in cancer screening.[37] Because HCC is rarely indolent as it had doubling times of 3–6 months in prior studies,[28,31] and only subjects asymptomatic at the time of HCC diagnosis were included in the analysis, this kind of bias is not likely to have been introduced in our study. Third, although most patients included in this study were infected by HBV, HBV itself would not have a negative effect on the ability of AFP to detect HCC as long as the HBV infection was not exacerbated, as actually in our case.[38]

In conclusion, we have demonstrated that measuring AFP to detect HCC can improve overall and cancer-specific mortalities in patients with chronic liver disorders, even after correcting for lead times due to AFP screening. This improvement may be due to detecting tumors at an earlier stage, so increasing the chance of curative treatment. Given the inherent

limitations of ultrasound and the lack of availability of other reliable markers, our data suggest that continuing to use the AFP assay together with established US examination in at-risk patients is clinically desirable and practically recommendable, supporting the relevant updated guidelines.

## Supporting information

**S1 Fig.** (A) overall mortality and (B) cancer-specific mortality of the AFP and AFP+US groups after propensity score matching in Model 1. Both overall and cancer-specific mortalities were significantly different in the two groups (Ps<0.001, respectively, by the log-rank test).
(DOCX)

**S2 Fig.** (A) Overall mortality and (B) cancer-specific mortality of the AFP and AFP+US groups after propensity score matching in Model 2. Neither overall nor cancer-specific mortal-ities were significantly different in the two groups (P = 0.200 and P = 0.230, respectively, by the log-rank test)
(DOCX)

**S3 Fig. Corrected survival of patients according to abnormal results of the HCC screening tools.** The differences between groups with respect to (A) overall mortality, and (B) cancer-specific mortality, were significant after adjusting for the calculated lead time (tumor doubling time 120 days, *P*<0.001 and *P*<0.001, respectively by the log-rank tests).
(DOCX)

**S4 Fig. Survival of patients according to abnormal results of the HCC screening tools in the cirrhotic subset.** The differences between groups with respect to (A) overall mortality, and (B) cancer-specific mortality, were significant (*P*<0.001 and *P*<0.001, respectively by the log-rank test).
(DOCX)

**S1 Table. Independent pre- and post-screening parameters related to survival in the entire HCC cohort (based on the AJCC system).**
(DOCX)

**S2 Table. Baseline characteristics of the AFP and AFP+US groups in the unmatched cohort and propensity score-matched cohorts using the variables included in Model 1 and Model 2, respectively.**
(DOCX)

**S3 Table. Survival analyses in the AFP group with adjustment for possible lead time (based on the BCLC system).**
(DOCX)

**S4 Table. Survival analyses in the AFP group with adjustment for possible lead time (based on the AJCC system).**
(DOCX)

**S5 Table. Independent pre- and post-screening parameters related to survival in the cir-rhotic subset (based on the BCLC system).**
(DOCX)

**S6 Table. Survival analyses in the AFP group with adjustment for possible lead time in cir-rhotic subset (based on the BCLC system).**
(DOCX)

## Acknowledgments

We thank Drs Danbi Lee, Kang Mo Kim, Young-Suk Lim, Han Chu Lee, Young-hwa Chung, and Yung Sang Lee, of the Department of Gastroenterology, Asan Medical Center, for data collection. None of these individuals received compensation for their work.

## Author Contributions

**Conceptualization:** Jihyun An, Ha Il Kim, Ju Hyun Shim.

**Data curation:** Jihyun An, Ha Il Kim, Seheon Chang, Ju Hyun Shim.

**Formal analysis:** Jihyun An, Ha Il Kim, Ju Hyun Shim.

**Supervision:** Ju Hyun Shim.

**Writing – original draft:** Jihyun An, Ha Il Kim, Ju Hyun Shim.

**Writing – review & editing:** Jihyun An, Ha Il Kim, Seheon Chang, Ju Hyun Shim.

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
