## [Decision Letter · Decision Letter 0]

22 Jun 2020

PONE-D-20-16448

Continued Value of the Serum Alpha-Fetoprotein Test in Surveilling At-Risk Populations for Hepatocellular Carcinoma

PLOS ONE

Dear Dr. JU HYUN Shim,

Thank you for submitting your manuscript to PLOS ONE. After careful consideration, we feel that it has merit but does not fully meet PLOS ONE’s publication criteria as it currently stands. Therefore, we invite you to submit a revised version of the manuscript that addresses the points raised during the review process.  The Reviewers and the Editorial team had major concerns that must. be addressed.

Please submit your revised manuscript within 60 days. If you will need more time than this to complete your revisions, please reply to this message or contact the journal office at plosone@plos.org. Please include the following items when submitting your revised manuscript:

We look forward to receiving your revised manuscript.

Kind regards,

Gianfranco D. Alpini

Academic Editor

PLOS ONE

Journal Requirements:

2. To comply with PLOS ONE submission guidelines, in your Methods section, please provide additional information regarding your statistical analyses. For more information on PLOS ONE's expectations for statistical reporting, please see https://journals.plos.org/plosone/s/submission-guidelines.#loc-statistical-reporting.

3. Please include your tables as part of your main manuscript and remove the individual files. Please note that supplementary tables (should remain/ be uploaded) as separate "supporting information" files

5. Your ethics statement must appear in the Methods section of your manuscript. If your ethics statement is written in any section besides the Methods, please move it to the Methods section and delete it from any other section. Please also ensure that your ethics statement is included in your manuscript, as the ethics section of your online submission will not be published alongside your manuscript.

Reviewers' comments:

Reviewer's Responses to Questions

**Comments to the Author**

1. Is the manuscript technically sound, and do the data support the conclusions?

Reviewer #1: Yes

Reviewer #2: No

Reviewer #3: Yes

2. Has the statistical analysis been performed appropriately and rigorously? 

Reviewer #1: Yes

Reviewer #2: No

Reviewer #3: Yes

3. Have the authors made all data underlying the findings in their manuscript fully available?

Reviewer #1: Yes

Reviewer #2: Yes

Reviewer #3: Yes

4. Is the manuscript presented in an intelligible fashion and written in standard English?

Reviewer #1: Yes

Reviewer #2: No

Reviewer #3: Yes

5. Review Comments to the Author

Reviewer #1: Jihyun An and coauthors in this study analysed hospital-based cancer registry, identifying 1,776 asymptomatic adults who were surveilled biannually with the AFP test and US and eventually diagnosed with HCC between 2007 and 2015. They have demonstrated that measuring AFP to detect HCC can improve overall and cancer-specific mortalities in patients with chronic liver disorders, even after correcting for lead times due to AFP screening. This improvement may be due to detecting tumors at an earlier stage, so increasing the chance of curative treatment.

The study is interesting and the message important.

I have only minor comments:

I wonder who are the 10-15% of patients under surveillance without chirrosis

It would be helpful for the reader to have a flowchart of case selection to figure out how you selected 1776 cases out of 9,615 patients

Reviewer #2: In this study, An et al. seek the potential of serum AFP testing for HCC. This study is poorly designed and it seems that the authors do not understand what they are doing and what they want to study.

• AFP test and US are not therapeutic treatments. It does not make sense to compare with AFP+US and calculate survival rates. If patients are positive for both AFP and US, it is highly likely that tumors are big enough to be detected by both methods. Therefore, the survival rate of AFP+US is low. It does not mean that AFP test is useful to detect HCC or AFP can provide better survival rates or patients should take AFP tests. The authors mention that AFP surveillance could be helpful, but data in this study do not prove anything. The authors do not understand what they are doing.

• Demographic characteristics should be similar between groups. For example, AFP group has more female patients that other groups, and some studies have shown that female patients have better survival rates than male patients. Age, smoking, alcohol consumption are related to survival rates. Since NAFLD and obesity are associated with cancer development and metastases, BMI also influences survival rates. These parameters are significantly different between groups, and then how can the authors compare survival rates? This study is poorly designed and no data provided in this manuscript are useful for readers.

Reviewer #3: Hepatocellular carcinoma (HCC) is a type of cancer with poor prognosis. The screening is critically important, with the commonly used methods of ultrasound and serum alpha-fetoprotein (AFP) test. The authors re-evaluated whether complimentary testing for serum AFP is helpful in surveilling for HCC in high-risk populations. The sample size of this study is large enough. The criteria for the enrollment and exclusion are reasonable. The statistics are sound. To further improve the manuscript, several comments were provided:

1. Since the study focuses on the continued value of AFP, it’s better to explain more about its original value in HCC screening. For example, in the part of introduction, the function of AFP and ultrasound screening should be explained more clearly. What are their advantages and disadvantages respectively? Moreover, what about their combinations?

2. There is something not very clear for the patient enrollment. It’s written that the study population consists of 9,615 patients, among which 5,786 are excluded because they did not undergo biannual regular surveillance tests for HCC. What kind of population are required to participate in this surveillance program? Are there any criteria?

6. PLOS authors have the option to publish the peer review history of their article (what does this mean?). If published, this will include your full peer review and any attached files.

Reviewer #1: No

Reviewer #2: No

Reviewer #3: No

---

## [Author Response · Author response to Decision Letter 0]

29 Jul 2020

Dear Editor,

We would like to thank the editors and reviewers for their careful consideration of our work. We believe that the suggestions made have strengthened our manuscript. A point-by-point response to the comments of the editors and reviewers follows. Please see the attached file "response letter to the reviewers." 

Comments of Reviewer 1

#1. I wonder who is the 10-15% of patients under surveillance without cirrhosis. It would be helpful for the reader to have a flowchart of case selection to figure out how you selected 1776 cases out of 9,615 patients

Reply: 

Most international guidelines advocate that, together with cirrhotic patients, non-cirrhotic patients with chronic hepatitis B, should undergo HCC surveillance, especially in the case of Asian populations (Heimbach JK et al., Hepatology 2018;67:358-380, Omata M et al., Hepatol Int 2017;11:317-370, Korean Liver Cancer Association and National Cancer Center, Gut and Liver, 2019;13:227-299) In line with these recommendations, most of the non-cirrhotic patients who were under surveillance were subjects with chronic hepatitis B (84.6%). In response to the reviewer’s comment, we have changed the flow chart of patient enrollment in Figure 1.

Comments of Reviewer 2

#1. AFP test and US are not therapeutic treatments. It does not make sense to compare with AFP+US and calculate survival rates. If patients are positive for both AFP and US, it is highly likely that tumors are big enough to be detected by both methods. Therefore, the survival rate of AFP+US is low. It does not mean that AFP test is useful to detect HCC or AFP can provide better survival rates or patients should take AFP tests. The authors mention that AFP surveillance could be helpful, but data in this study do not prove anything. The authors do not understand what they are doing.

Reply:

We thank the reviewer for this helpful comment. First, as the reviewer mentioned, the AFP test and US are not therapeutic treatments but surveillance tools for detecting HCC, and the most important aim of surveillance is to reduce deaths from the target disease (Black WC et al., J Natl Cancer Inst 2002;94:167-73, Tabar L et al., J Med Screen 2002;9:159-62). Disease-specific mortality and all-cause mortality are used extensively as suitable end points in trials of cancer screening (Prasad V et al., BMJ 2016;352:h6080, Saquib N et al. Int J Epidemiol 2015;44:264-77). A randomized trial and several cohort studies have also demonstrated an association between HCC surveillance and early tumor detection and improved survival in patients with cirrhosis (Costentin CE et al., Gastroenterology 2018;155:431–442, Meer SV et al., Journal of Hepatology 2015;63:1156-1163, Zhang BH et al., J Cancer Res Clin Oncol 2004;130:417–422).

Second, as the reviewer suggested, the tumors detected by both AFP and US were significantly larger than those detected by AFP alone or US alone in our series. However, survival outcomes were not different between the AFP and AFP+US groups after multivariate analysis including tumor stage and anti-cancer treatment (i.e., model 2). These results were confirmed by a propensity score matching analysis, which has been added in response also to comment #2 of reviewer 2. This means that the prognoses of patients with HCC at the same stages would be similar regardless of the surveillance tests by which lesions were detected. Therefore, we speculated that AFP testing could play a worthwhile role in the early detection of HCC, complementing the limitations of US screening and improving survival by providing the opportunity for curative treatments.

#2. Demographic characteristics should be similar between groups. For example, AFP group has more female patients that other groups, and some studies have shown that female patients have better survival rates than male patients. Age, smoking, alcohol consumption are related to survival rates. Since NAFLD and obesity are associated with cancer development and metastases, BMI also influences survival rates. These parameters are significantly different between groups, and then how can the authors compare survival rates? This study is poorly designed and no data provided in this manuscript are useful for readers.

Reply: 

To overcome the retrospective design of our study in which demographic characteristics were not similar across the groups, we used a multivariate regression method including all relevant covariates related to the outcomes of interest. Furthermore, in response to the reviewer’s comment, we matched the AFP group with the AFP+US group for all confounding baseline variables in models 1 and 2 using a rigorous matching method based on propensity scores, and obtained similar results to the original non-matched analyses. We have added these findings in the Methods, Results and Discussion sections, Supplementary Table 2, and Supplementary Figures 2-3 as follows:

(Methods) To reduce the impact of potential confounding effects in the AFP group and the AFP+US group, rigorous adjustment was made for significant differences in baseline characteristics in Model 1 and Model 2 by propensity score-based matching. Propensity scores were matched for the two groups based on differences of ±0.05 in the scores. Differences in overall mortality and cancer-specific mortality between the matched groups were compared using Cox regression models, with robust standard errors that accounted for the clustering of matched pairs.[24] A P<0.05 was considered statistically significant. All statistical analyses were performed using R package MatchIT.[25]

(Results) In addition, these results were recapitulated after PS matching between the AFP and AFP+US groups (S2 Table and S1-S2 Fig.).

(Discussion) Because of the finding that this difference in survival between the patients identified by AFP with and without abnormal US findings was maintained after correcting for lead time, combined with confounding adjustment, we suggest that the observed survival benefit was due to downward stage migration followed by more frequent receipt of potentially curative treatment. 

Comments of Reviewer 3

#1. Since the study focuses on the continued value of AFP, it’s better to explain more about its original value in HCC screening. For example, in the part of introduction, the function of AFP and ultrasound screening should be explained more clearly. What are their advantages and disadvantages respectively? Moreover, what about their combinations?

Reply:

Thank you for your helpful comment. We have added the more relevant content to the Introduction section of the revised manuscript as follows:

(2nd paragraph of Introduction) Surveillance for HCC is thought to be a way to detect lesions at an earlier stage and improve clinical outcomes in asymptomatic at-risk populations.[4, 5] Currently, ultrasonography (US) is regarded as the backbone of screening for HCC, but it has practical limitations in terms of high operator-dependency and variable sensitivity. A recent meta-analysis highlights the suboptimal (<50%) sensitivity of US for detection of HCC at an early stage, although the sensitivity and specificity of US in detecting HCC of any stage exceed 90% (Tzartzeva K et al., Gastroenterology 2018;154:1706–1718). The serum alpha-fetoprotein (AFP) assay is the only serological screening tool for early detection of HCC that reliably meets the final 5-phases criterion for possibly reducing the population disease burden listed by the Early Detection Research Network (EDRN), an initiative of the National Cancer Institute (NCI).[6] However, the use of AFP in surveillance is subject to ongoing debate, even as an adjunct to USG, due to issues about cost-effectiveness.[7-9] There have been false-positive results for HCC detection due to the AFP elevation encountered in chronic liver disease, and false-negative results in HCCs not secreting AFP (Wong RJ et al., Clin Liver Dis 2015;19:309–323). Unfortunately, there are no robust or promising next-generation biomarkers available for clinical use in screening or diagnostic systems

Both the American Association for the Study of Liver Diseases (AASLD) and the National Comprehensive Cancer Network (NCCN) guidelines for HCC, newly updated in 2017, re-introduce serum AFP testing as an additional surveillance method because of its potential benefit for patients with HCCs secreting AFP (Heimbach JK et al., Hepatology 2018;67:358-380, Benson AB el al, J Natl Compr Canc Netw 2017;15:563-573). Recent interesting efforts in the U.S. to develop a computer model for detecting HCC also included serum AFP values in the prediction calculator (El-Serag HB, Gastroenterology 2014;146:1249-1255, White DL, Gastroenterology 2015;149:1986-1987).

#2. There is something not very clear for the patient enrollment. It’s written that the study population consists of 9,615 patients, among which 5,786 are excluded because they did not undergo biannual regular surveillance tests for HCC. What kind of population are required to participate in this surveillance program? Are there any criteria?

Reply:

The practice guidelines of the Korean Liver Cancer Society for the management of HCC recommend that surveillance for HCC should be performed with liver US plus serum AFP measurement every 6 months in high-risk subjects (i.e., patients with chronic hepatitis B, chronic hepatitis C, and liver cirrhosis) (Korean Liver Cancer Association and National Cancer Center, Gut and Liver, 2019;13:227-299). However, the HCC surveillance rate in real-world practice is quite low. A Korean tertiary hospital cohort study found that the previous surveillance rate was only 39.8% in newly diagnosed HCC patients, and adherence to the strict 6-monthly surveillance rate of the current guidelines was only 21.9%, suggesting a suboptimal surveillance status (Im S et al., Cancer Res Treat 2019;51:1357-1369). Routine surveillance also occurred in only 12.0% of the HCV-infected population, based on U.S. Veterans Health Administration data (Davila JA et al., Ann Intern Med. 2011;154:85–93) Since the objective of the study was to evaluate the effect of surveillance of serum AFP on a background of US examination, we specifically included asymptomatic patients who underwent standard surveillance for HCC at 6-monthly intervals.

---

## [Decision Letter · Decision Letter 1]

10 Aug 2020

Continued Value of the Serum Alpha-Fetoprotein Test in Surveilling At-Risk Populations for Hepatocellular Carcinoma

PONE-D-20-16448R1

Dear Dr. Shim,

We’re pleased to inform you that your manuscript has been judged scientifically suitable for publication and will be formally accepted for publication once it meets all outstanding technical requirements.

Kind regards,

Gianfranco D. Alpini

Academic Editor

PLOS ONE

Additional Editor Comments (optional):

Reviewers' comments:

Reviewer's Responses to Questions

**Comments to the Author**

1. If the authors have adequately addressed your comments raised in a previous round of review and you feel that this manuscript is now acceptable for publication, you may indicate that here to bypass the “Comments to the Author” section, enter your conflict of interest statement in the “Confidential to Editor” section, and submit your "Accept" recommendation.

Reviewer #2: All comments have been addressed

Reviewer #3: All comments have been addressed

2. Is the manuscript technically sound, and do the data support the conclusions?

Reviewer #2: Yes

Reviewer #3: Yes

3. Has the statistical analysis been performed appropriately and rigorously? 

Reviewer #2: Yes

Reviewer #3: Yes

4. Have the authors made all data underlying the findings in their manuscript fully available?

Reviewer #2: Yes

Reviewer #3: Yes

5. Is the manuscript presented in an intelligible fashion and written in standard English?

Reviewer #2: Yes

Reviewer #3: Yes

6. Review Comments to the Author

Reviewer #2: An et al. seek the potential of serum AFP testing for HCC. I have no further comments. I have no competing interests.

Reviewer #3: (No Response)

7. PLOS authors have the option to publish the peer review history of their article (what does this mean?). If published, this will include your full peer review and any attached files.

Reviewer #2: No

Reviewer #3: No

---

## [Editor Report · Acceptance letter]

14 Aug 2020

PONE-D-20-16448R1 

Continued Value of the Serum Alpha-Fetoprotein Test in Surveilling At-Risk Populations for Hepatocellular Carcinoma 

Dear Dr. Shim:

I'm pleased to inform you that your manuscript has been deemed suitable for publication in PLOS ONE. Congratulations! Your manuscript is now with our production department. 

Kind regards, 

on behalf of

Dr. Gianfranco D. Alpini 

Academic Editor

PLOS ONE